# “Grafting-To” Covalent Binding of Plasmonic Nanoparticles onto Silica WGM Microresonators: Mechanically Robust Single-Molecule Sensors and Determination of Activation Energies from Single-Particle Events

**DOI:** 10.3390/s23073455

**Published:** 2023-03-25

**Authors:** Mariana P. Serrano, Sivaraman Subramanian, Catalina von Bilderling, Matías Rafti, Frank Vollmer

**Affiliations:** 1INIFTA-CONICET, Departamento de Química, Facultad de Ciencias Exactas, Universidad Nacional de La Plata, La Plata B1900, Argentina; 2Living Systems Institute, Department of Physics & Astronomy, University of Exeter, Exeter EX4 4QD, UK

**Keywords:** plasmonic nanoparticles (PNPs), gold nanorods (GNr), silanization reaction, WGM microresonators

## Abstract

We hereby present a novel “grafting-to”-like approach for the covalent attachment of plasmonic nanoparticles (PNPs) onto whispering gallery mode (WGM) silica microresonators. Mechanically stable optoplasmonic microresonators were employed for sensing single-particle and single-molecule interactions in real time, allowing for the differentiation between binding and non-binding events. An approximated value of the activation energy for the silanization reaction occurring during the “grafting-to” approach was obtained using the Arrhenius equation; the results agree with available values from both bulk experiments and ab initio calculations. The “grafting-to” method combined with the functionalization of the plasmonic nanoparticle with appropriate receptors, such as single-stranded DNA, provides a robust platform for probing specific single-molecule interactions under biologically relevant conditions.

## 1. Introduction

Recently emerged sensing technologies based on optical whispering gallery mode (WGM) resonators have proved useful for studies of the interactions between nanoparticles and molecules, even under challenging experimental conditions in aqueous buffers and when using complex biological samples [1,2,3]. In WGM sensing, the changes in the refractive index of the medium surrounding the sensor are detected from the frequency or linewidth shifts of the WGM resonances [4]. WGMs are resonances with a very high Q factor (ranging 10^5^–10^7^) in which the Q factor is defined as the ratio of the WGM resonance frequency divided by its linewidth. High-Q WGM resonators enable the most sensitive nanoparticle/molecule detection and can be straightforwardly fabricated from the controlled melting of commercial fused silica optical fibers using, for example, a CO_2_ laser [5]. The sensitive detection of the refractive index (RI) perturbations of WGM modes enables many different sensing applications. For example, perturbations caused by temperature, pH, or ionic strength variations were detected from WGM resonance frequency shifts [6,7,8,9]. Detection down to the single-molecule level has been made possible through enhancing the WGM signal by attaching plasmonic nanoparticles (PNPs) to a ~100 μm diameter glass microsphere WGM resonator [10,11,12,13,14]. The integration of PNPs with WGMs, resulting in so-called optoplasmonic WGM sensors, led to major breakthroughs in biosensing; for example, the real-time detection of the single-molecule hybridization events of DNA oligonucleotides and the detection of single-molecule WGM signals corresponding to disulfide bond formation [3,15,16,17,18]. PNPs with different geometries, such as nanorods, nanostars, and nanoshells, can be employed for assembling a single-molecule optoplasmonic WGM sensor [4]. Gold nanorods (Au-NRs) have been used in many of the WGM sensing applications because they are commercially available, provide a strong optical near-field intensity enhancement (the typical enhancement factor at the tip of the nanorod is calculated to be approx. 800), and a wide range of plasmon resonance wavelengths, depending on the nanorod aspect ratio [19,20,21,22,23].

Reproducible and versatile protocols for the surface functionalization of WGMs for the permanent attachment of Au-NR to WGM glass microresonators are necessary in order to yield stable optoplasmonic WGM sensors capable of molecule sensing under various challenging experimental conditions. Self-assembly methods in aqueous solutions of PNPs are particularly useful for fabricating optoplasmonic WGM sensors because, owing to their simplicity, sophisticated equipment is not required. Furthermore, they allow for the real-time sensing of the exact number of attached PNPs.

Methods for optoplasmonic WGM sensor surface functionalization and assembly from solution can be grouped into two main categories, namely, (i) methods that use physico-chemically modified glass resonators for the subsequent PNP attachment through non-covalent interactions, e.g., electrostatic binding or layer-by-layer (LbL) polymer-assisted assembly [24,25]; (ii) methods that employ surface functionalization to introduce chemical moieties for the subsequent covalent attachment of the PNPs to the glass microresonators, e.g., through surface silanization reactions [26]. Attention must be paid to methods based on surface silanization because slight changes in reaction conditions yield striking differences in the chemistry and morphology of the resulting silane layer [27]. For example, silanization with 3-aminopropyltriethoxysilane (APTES) can produce either smooth and thin monolayers, exposing -NH_2_ moieties ideally suited for PNP attachment, or thick, non-smooth self-polymerized multilayers which are less desirable in WGM-related applications because of the possible degradation of the Q factor due to surface scattering [28,29]. We will briefly outline the two most common silanization methods employed in the grafting-to approach for PNPs.

Non-covalent PNP attachment by electrostatic interactions. Usually, procedures for the non-covalent attachment of PNPs onto glass microresonators employ surface positioned -NH_2_ moieties introduced in a silanization step with APTES. The pH-dependent electrostatic charge of primary amine moieties is used for the unspecific electrostatic attachment of PNPs bearing the opposite charge, such as citrate-capped PNPs [8]. Single PNP binding events can be detected from the observed wavelength/linewidth WGM signal shifts [19]. The optoplasmonic WGM sensors thus assembled were shown to be suitable for the detection of diverse analytes in aqueous buffer solutions, including single ions, DNA, and polymerase proteins [9,30]. Problems appeared when dealing with experimental conditions requiring, e.g., the use of buffers with pH > 10.5 or increased ionic strength for which interactions between PNPs and microresonator surface weaken, thus affecting the stability of the PNP on the sensor and resulting in an increase in the baseline/background WGM shift signals [8].

Covalent PNP attachment by silanization. The previously discussed shortcomings of electrostatic PNP binding methods can be circumvented by resorting to covalent PNP attachment. Covalent binding methods for PNPs have been used in other single-molecule techniques based on AFM and optical tweezers.

Here, we introduce a covalent method for PNP attachment by silanization for its use in WGM sensing and propose a reliable protocol which yields mechanically robust sensors modified with PNPs [31]. The method is based on a two-step approach (see Figure 1). Step one consists of the modification of PNPs with MPTMS (3-mercaptopropyl-trimethoxysilane), which replaces stabilizing agents such as CTAB/citrate through ligand exchange reactions [32]. The ligand exchange, directed by high-affinity thiol–Au interactions, yields the surface positioning of silanizable moieties due to the formation of thiolate self-assembled monolayers (SAMs) of MPTMS; the obtained PNPs will hereafter be referred to as Au-NR@HS-Si(OH)_n_ [33,34,35,36]. In step two, a controlled silanization reaction resembling the “grafting-to” approach widely employed in polymer science [37,38] generates in the in situ covalent attachment of Au-NR@HS-Si(OH)_n_ to silica WGM microresonators (see Figure 1).

It is noteworthy to mention that by controlling the reaction conditions for the above-described reaction, i.e., the concentration of PNPs, pH, and ionic strength, it is possible to optimize the covalent attachment. The covalent attachment of PNP is observed from a step-like signal in the WGM sensor (wavelength-shift) time traces, whereas spike-like signals indicate unsuccessful, transient PNP–microsphere interactions. The real-time WGM sensor signals can then be then for estimating values for the silanization activation energy corresponding to a single-molecule silanization reaction by applying the empirical Arrhenius equation. Thus far, reported estimates for the process were based on bulk measurements or ab initio calculations [39,40]. In addition, specific single-molecule sensing by optoplasmonic WGM sensors often requires the attachment of a biorecognition element to the Au-NR. We demonstrate this for the grafting-to approach by attaching thiolated 21mer DNA oligonucleotides and demonstrating single-molecule sensing of DNA hybridization. 

## 2. Materials and Methods

### 2.1. Chemicals 

Citrate-stabilized gold nanorods (NanoXact AuNR) with dimensions of 17 nm × 47 nm and a plasmonic wavelength centered at 660 nm, nearly matching the wavelength of the Toptica DL Pro 642 nm laser used for WGM sensing, were used as provided by nanoComposix (San Diego, CA, USA); see the Appendix A for further characterization details. An MPTMS silanization agent (3-mercaptopropyltrimethoxysilane), was employed as provided by Sigma-Aldrich (St. Louis, MO, USA). Common chemicals used for the regulation of pH and ionic strength were also purchased from Sigma-Aldrich: Sodium Chloride, Sodium Citrate, and Sodium Hydroxide. Absolute ethanol was provided by Sigma-Aldrich, and milli-Q water was obtained from a Thermo-Fisher Scientific Inc. filtering system. Alkaline piranha stock solution was prepared by mixing ammonium hydroxide (NH_4_OH, 28% water solution, Sigma Aldrich) and hydrogen peroxide (H_2_O_2_, 30% Sigma Aldrich) in a (3:1) volume proportion. HEPES buffer (N-(2-Hydroxyethyl)piperazine-N′-(2-ethanesulfonic acid)) and Tris-Carboxy Ethyl Phosphene (TCEP) were provided by Sigma-Aldrich and used as buffer and reducing agent for breaking disulfide bonds in single-stranded oligonucleotide DNA (SS-A9-DNA: 5′-SH-AAAAAAAAAGTTATGTTATGT-3′). Both SS-A9-DNA and the full complementary (C12: 5′-ACATAACATAAC-3′) were obtained from Eurofins Scientific (see Appendix A).

### 2.2. Colloidal Modification of Citrate-Capped Au-NR Using MPTMS

Stock 0.25% *v*/*v* ethanolic MPTMS solutions were prepared from pure MPTMS (δ = 1.050 g mL^−1^, 95% *w*/*w*) and stored in dry and cool conditions for their subsequent use in the reactions with PNPs. In particular, the hereby employed citrate-capped AuNRs bear special interest because most PNP functionalization protocols are designed for this highly biocompatible stabilizing agent. Therefore, it should be mentioned for the sake of completeness that as many nanorod synthesis procedures result in cetyltrimethylammonium bromide- or CTAB-coated PNPs, a recently published protocol for the ligand exchange between CTAB and citrate was tested and proven suitable, thus enabling the use of CTAB-capped PNPs with our “grafting-to” approach [32]. The procedure that followed the preparation of 100 μL of MPTMS-modified AuNRs concentration 0.5 OD, hereafter referred to as Au-NR@HS-Si(OH)_n_, was as follows: 48 μL of milli-Q water was added to 50 μL citrate-capped AuNRs (NanoXact, concentration 1.0 OD); after brief mixing with an ultrasonic bath, 2 μL of MPTMS solution (a (1:10) dilution in milli-Q water of MPTMS stock) was introduced dropwise under magnetic stirring and left to react for 40 min. The procedure yielded a final ≈1:1000 molar ratio of AuNR: MPTMS, much lower than compared to the reported concentration ratios employed for production of thick silica coatings in synthesis protocols for AuNR@silica core–shell composites [33]. The conditions described are known to cause the partial hydrolysis of MPTMS ethoxy groups, thus yielding exposed silanol moieties on the Au-NR surface. This can be rationalized by considering the oxidative chemisorption occurring upon Au–thiol interactions, which replace the relatively weakly-bound citrate capping [34]. The fact that Au-NR@HS-Si(OH)_n_ remain well-dispersed and that the above-described 660 nm plasmon absorption band was preserved (see Appendix A) suggest the successful surface positioning of hydrophilic silanol moieties. All experiments were conducted under neutral pH conditions or a pH = 3 citrate buffer (see Appendix A). However, care must be taken in this regard, e.g., it is known that strongly alkaline conditions (pH ≈ 10) and the presence of ammonia catalyze condensation reactions lead to the uncontrolled formation of a thick silica shell, which would not be suitable for the herein intended use in combination with WGM resonators.

### 2.3. Colloidal Hybridization of Au-NR with MPTMS and Single-Strand A9 DNA

A detailed description of the standard procedure employed for the modification of PNPs with SS-A9-DNA is provided in the Appendix A. Briefly, the protocol included the following steps: (i) the activation of single-strand, thiol-baring SH-A9 oligonucleotide DNA with TCEP (TCEP:A9) at a molar ratio of (50:1). (ii) Au-NR functionalization using activated A9-DNA with molar proportions (A9-DNA:Au- NR) of (1:1500), followed by mixing in an ultrasonic bath at 40 °C for 60 min. This step allows for the surface positioning of thiol-terminated activated A9-DNA. (iii) The addition of 500 mM citrate buffer at pH = 3 for 30 min to the DNA-hybridized nanorods to obtain (A9-DNA-Au-NR).

### 2.4. WGM Experimental Setup

The optoplasmonic WGM microsensor setup used for the experiments of the Au-NR covalent is detailed in the Appendix A [19]. Briefly, silica microspheres with an average diameter ≈80–90 μm were fabricated with a CO_2_ laser glass-melting setup. The as-produced microsphere was then mounted on the prism-based setup in which WGM modes were excited using a tunable external laser (Toptica DL Pro, 642 nm) via the prism coupling. The sample cell in which the sensor was immersed consisted of a 300 μL volume polydimethylsiloxane (PDMS) gasket in which all described in situ procedures were performed. 

### 2.5. WGM Data Processing and Determination of Independent Events

The WGM spectra were acquired by scanning the external cavity diode laser over a small bandwidth around the resonance at a scan rate of 50 Hz. The resonance wavelength and linewidth were extracted from the recorded spectra using a custom Labview (National Instruments Inc., Austin, TX, USA) code [14]. The time traces of WGM resonance position (λ) and linewidth (κ) were then analyzed using a custom MATLAB program (Mathworks Inc., Natick, MA, USA) to determine peaks and steps in the recorded time series arising from single nanoparticle interaction events. The details of peak detection method are described in the Appendix A. The main parameters extracted were, namely, the time of the occurrence of an event (*t*), event amplitude (*h*), and event duration (*τ*).

### 2.6. AFM Experiments and Preparation of Ex-Situ Samples

Additional ex situ experiments with Atomic Force Microscopy (AFM) were carried out using silica surfaces functionalized following a procedure identical to the above-described procedure for WGM microresonators. Silica glass was cleaned and then exposed to Au-NR@HS-Si(OH)_n_ PNPs. After appropriate exposure periods, the substrates were washed abundantly with alkaline solutions to remove non-covalently attached PNPs and then dried in an inert atmosphere. 

## 3. Results and Discussion

### 3.1. Surface Modification of Silica WGM Microspheres and the Effect of Alkaline Piranha Solution Treatment on the Interaction with PNPs

As-prepared silica microresonators feature silanol moieties, conferring either positive (-Si-OH_2_^+^), neutral (-Si-OH), or negative (-Si-O^−^) surface charges, depending on the pH of surrounding media; the silica isoelectric point (IEP) pH is ≈2–3 [41]. Activation procedures with piranha solutions result in a higher surface density of the silanol moieties but do not modify IEPs; instead, they cause a surface charge density increase due to the additional silanol moieties created (see Figure 2a) [42]. Bearing in mind that citrate-capped AuNRs remain negatively charged for a wide pH range, the electrostatic affinity of PNPs towards the microresonator surface can be modulated using pH [34]. For example, an acidic pH produces attractive electrostatic interactions and thus a surface charge suitable for the non-specific attachment of PNPs. However, such attachment is not always robust, and the stability of the assembly could be compromised when modifying the working pH or ionic strength (e.g., for biologically relevant applications requiring near-neutral mild conditions). Therefore, a method for the covalent bonding of PNPs is developed here by employing surface-positioned silanizable moieties enabling condensation reactions with the glass silanol groups of the microsphere.

We first performed experiments aimed at understanding the effect of the surface concentration of silanol moieties of the glass microsphere on the interaction with Au-NR@HS-Si(OH)_n_. To this end, the event rate of WGM-PNP interaction events was determined from the WGM signal traces, measured with the as-prepared WGM (see Figure 2b), compared to the alkaline-piranha-treated WGM microspheres (see Figure 2c). Non-permanent interactions, signified by the spike-like events in the WGM sensor traces, were observed for both conditions (the PNPs’ picomolar concentration and neutral pH). Spike events due to the transient interactions of the PNPs with the glass microsphere are seen in the WGM wavelength and linewidth shifts. Notably, there was a marked increase of such event rates (1.4 s^−1^) when using piranha-treated microresonators, ascribable to the increment in surface silanol moieties (see Figure 2b,c).

Having established the effect of the piranha treatment and the role of silanol moieties in affecting interaction with PNPs, we selected non-treated WGM resonators for further use. Avoiding the piranha treatment brings a much-desired simplification of the protocol which, if successful, would provide better reproducibility. Furthermore, piranha treatment causes an increased rugosity on the surface which may affect the maximum reachable Q factor.

Next, we analyzed the Δκ and Δλ shift signals for the non-treated WGM microspheres further interacting with the Au-NR@HS-Si(OH)_n_. Figure 3 shows the linear dependence of determined event rate vs. the PNP concentration, both for linewidth and wavelength shifts (event rates were averaged over 100 s, and increasingly higher concentrations of Au-NR@HS-Si(OH)_n_, namely, 9, 14, 18, and 28 pM for aqueous solutions with pH = 7, were used). Figure 4 shows the histograms for the waiting times (time intervals between spike-like signals) and their Poissonian distribution for the linewidth traces (see Appendix A). The observation of both the concentration linear dependence and the Poissonian distribution of resonator–PNP interaction events supports the hypothesis that under the conditions employed, the time series obtained can be modeled assuming single-particle free diffusion. The linewidth shift signals were proportional to the polarizability squared of the particle, whereas wavelength shift signals were proportional to the polarizability of the particle. When larger particles and plasmonic particles are considered, this means that the wavelength shifts often provide a larger signal to noise ratio (SNR) over the linewidth shifts. In practice, the magnitude of the single-particle shift signals varies because the nanorods can be oriented arbitrarily and at random positions on the microsphere. In the experiments shown in Figure 3, the linewidth shift traces were able to record more of the single-particle events. The difference in the SNR is the reason why more events (an approx. 10% higher rate) were recorded for linewidth shift traces vs. wavelength shift traces. Since not all single particle events were detected, it is expected that the calculated spike rate in Figure 3 should be larger than the one that is measured. 

Considering free-diffusing particles subjected to Brownian motion as a valid approximation for the PNPs, an event rate value can be derived employing the Stokes–Einstein equation, Fick’s law, and an approximate value for the active sensing area on the microresonator, as determined by the lateral profile of WGM (see Appendix A for further details). The comparison between the experimental results and such calculations for increasingly higher concentrations of PNPs is shown in Figure 5. Although the deviation in absolute frequency values is considerable, the trend obtained from such simplistic model is adequate and accurately captures experimental features. There are a number of reasons why the calculated and experimental rates of interaction events are not expected to match numerically; namely, the exact mode used for sensing has not been identified, the sensing area is an approximate value, the electrostatic interactions between both PNPs and PNPs/microresonator surface are highly dependent on the (not straightforwardly measurable) surface charge magnitude, and the presence of an electrical double-layer, which disturbs fields and depends on the ionic strength and concentration of the species present. 

### 3.2. Determination of Regimes Leading to Permanent and/or Transient Interaction Events between PNPs and WGM Microresonator

For Au-NR@HS-Si(OH)_n_, both electrostatic and covalent interactions are possible, while they are not possible for the citrate-capped Au-NR covalent binding. To illustrate this point, Figure 6a shows WGM sensor traces corresponding to the citrate-capped PNPs; the experiment shows only spike events (pH 7, NaCl 10 mM, and 6 pM Au-NR). The same conditions were used in experiments with the PNPs exposing silanol moieties and Au-NR@HS-Si(OH)_n_: only steps events are observed (Figure 6b).

Figure 6c shows the time series for conditions in which both spikes and steps occur, namely at 10 mM sodium citrate buffer, pH = 3, and an Au-NR@HS-Si(OH)_n_ concentration of 2 pM. As is also shown in Figure 6d, there is a direct dependence of the event rate on the ionic strength for a fixed PNP concentration. Bearing in mind that constant pH values ensure a comparable surface charge (and thus electrostatic interactions), the observed increment in both event rates for an increasingly higher ionic strength can only be rationalized by taking into account Debye screening-length reduction.

Figure 7 shows the observed changes in Q factor values for the microresonators exposed to PNPs capable of binding either electrostatically or covalently upon pH variations. The Q factor is calculated as the ratio of the resonance wavelength (642 nm) divided by the spectral linewidth (FWHM) of the attenuation peak (Δκ). For covalently attached PNPs, it was observed that increasing the pH after the modification step at pH = 3 does not affect the Q factor magnitude, which suggests the robustness of the surface positioning method. However, this was not the case when using non-specific electrostatic PNP attachment (see Appendix A for control experiments carried with citrate-capped Au-NRs). 

Further support to our working hypothesis (i.e., silanization reactions result in a covalent attachment that is visible as step events) was obtained by carrying out additional ex situ experiments with Atomic Force Microscopy (AFM). Silica glass was cleaned and then exposed to Au-NR@HS-Si(OH)_n_ PNPs. Figure 8 shows the AFM images obtained, which evidence the successful covalent PNP bonding attained.

### 3.3. Silanization Activation Energy Calculations via Arrhenius Equation

The PNP attachment achieved through the silanization reaction can be assumed to proceed in a single elementary step, a simplistic approach already employed with success in the formulation of mechanisms for heterogeneous catalytic reactions under ultra-high vacuum (UHV) conditions [43,44]. The formation of a single Si-O-Si bond is assumed to be sufficient for the attachment of a PNP (Au-NR@HS-Si(OH)_n_), and results in the appearance of a single step-like WGM sensor signal. A simplified rate law corresponding to the silanization reaction can then be written, as in Equation (1):(1)$${Rate}_{Si-O-Si}=\frac{d\theta_{Si-O-Si}}{dt}=k\times \theta_{WGM-Si-OH}\times C_{PNP-Si-OH}$$
where the silanization rate (${Rate}_{Si-O-Si}$) depends on the reaction constant (k), the surface coverage of silanol moieties in the microresonator ($\theta_{WGM-Si-OH}$), and the concentration of silanized PNPs ($C_{PNP-Si-OH}$). Bearing in mind that the measured step event rate represents the effective reaction rate (${Rate}_{Si-O-Si}$ = 0.01185 events/s), and also given that for the conditions used (very low, 2 pM PNP concentration, corresponding to Figure 6d), the $\theta_{WGM-Si-OH}$ remains virtually constant, the following expression for the apparent reaction rate constant ($k^{\prime }$) can be derived (Equations (2) and (3)):(2)$$k^{\prime }=k\times \theta_{WGM-Si-OH}=\frac{{Rate}_{Si-O-Si}}{C_{PNP-SI-OH}}$$
(3)$$k^{\prime }=1.185\times 10^{-14}\;M^{-1}s^{-1}$$

Within the framework of such an approximation, the determined total event rate (spikes plus steps, 0.9 events/s, see Appendix A, for the corresponding histograms) would represent the Arrhenius frequency factor (*A*), and thus the activation energy (*E_a_*) can be obtained from the expression for the apparent reaction rate constant at the considered temperature (298 K) (Equations (4) and (5)):(4)$$k^{\prime }=Ae^{-\frac{E_{a}}{RT}}$$
(5)$$E_{a}=80.80\;kJ\;{mol}^{-1}$$

Despite the simplification, the activation energy value obtained is remarkably close to what is reported in the literature from ab initio calculations (approx. 100 kJ mol^−1^). It is worth mentioning that aside from the commonly accepted constraints of calculation methods (i.e., slab size limitations due to computational power available), the theoretical estimations suffer from overshooting, which arises from the impossibility of decoupling simultaneous processes such as adsorption or water coordination, which account nicely for the differences observed between the present work and the previously reported values [39,40].

For Au-NR@HS-Si(OH)_n_, both electrostatic and covalent interactions are possible, while the citrate as-prepared silica microresonators feature silanol moieties which confer either a positive (-Si-OH_2_^+^), neutral (-Si-OH) or negative (-Si-O^−^) surface charge, depending on the pH of surrounding media: the silica-based setup in which WGM modes were excited using a tunable external laser (Toptica DL Pro, 642 nm) via prism coupling. The sample cell in which the sensor was immersed consisted of a 300 μL volume polydimethylsiloxane (PDMS) gasket in which all described in situ procedures were performed. 

### 3.4. Complementary Oligonucleotide Detection Using Covalently Attached

We modified the Au-NR@HS-Si(OH)_n_ with a 21mer DNA oligonucleotide (see Appendix A) prior to the grafting-to approach to demonstrate the suitability of the assembled sensor in single-molecule DNA detection through hybridization to C12 complementary oligonucleotides. Figure 8 shows the WGM signal traces corresponding to baseline (before adding C12) and after the addition of 166 nM C12 solution, toghether with the linear dependence for the spike event rate and histograms corresponding to more than 300 events extracted from the time series (see Appendix A). From a careful analysis of the time series obtained, event rates were calculated (the conditions employed 100 mM NaCl and 10 mM HEPES buffer at pH = 7.4). The control experiments were carried out using a non-complementary DNA sequence and no interactions were detected for the same conditions employed above (see Appendix A). The determined rate increased linearly with the DNA concentration, as shown in Figure 9c, as would be expected for single-molecule events. Results support the single-molecule nature of the process, and we can extract an average kinetic on-rate (k_on_) for the full complementary 12mer obtained from the linear fitting of the data, which yields a k_on_ = 1.05 × 10^5^ M^−1^s^−1^. Considering 21 receptors, R^2^ = 0.995 (see Appendix A). This is in agreement with already published values for closely related model systems [14].

## 4. Conclusions

The use of non-covalent interactions for surface modification and even inclusion in layer-by-layer film architectures of plasmonic nanoparticles has been explored extensively and has proven to be a very versatile approach (enabling single-molecule detection limit [3] and, e.g., very sensitive reusable biosensors with immunologically conferred specificity towards bacteria cells [45]). Our contribution is centered in exploring a novel and feasible procedure for the covalent attachment of PNPs onto silica surfaces, which was successfully employed for the modification of WGM microresonators, thus adding a new degree of robustness and reproducibility to a highly sensitive technique. The proposed “grafting-to” approach enables several interesting options, such as the possibility of including core@shell PNPs with, e.g., tailored porosity and affinity provided by silica coatings. The concept explored herein enables the detailed monitoring of interaction events between modified PNPs and microresonators. It was shown that WGM microresonators can be successfully employed for the evaluation of activation energies at the single-molecule level under challenging conditions such as room temperature and aqueous environments. The robustness attained through the silanizable moieties positioned on the PNPs does not hinder the possibility of monitoring nucleic acid interactions for biosensing. Moreover, we envision that the method reported could be used to extend the scope of recent work carried on the study of temperature effect on enzymatic activity at the single-molecule level, allowing access to pH and ionic strength conditions that do not yield stable electrostatically assembled Au-NR-WGM sensors [46].

## Figures and Tables

**Figure 1 sensors-23-03455-f001:**
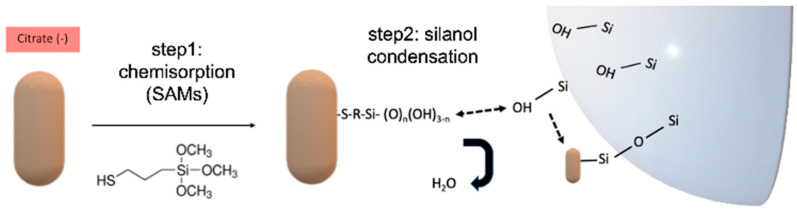
Sequential modification of citrate stabilized Au-NR using a two-step procedure. Step 1, Au-S chemisorption on the PNPs; Step 2, condensation reactions between silanol moieties present on the microresonator surface and MPTMS (3-mercaptopropyl-trimethoxysilane)-modified Au-NR (Au-NR@HS-Si(OH)_n_).

**Figure 2 sensors-23-03455-f002:**
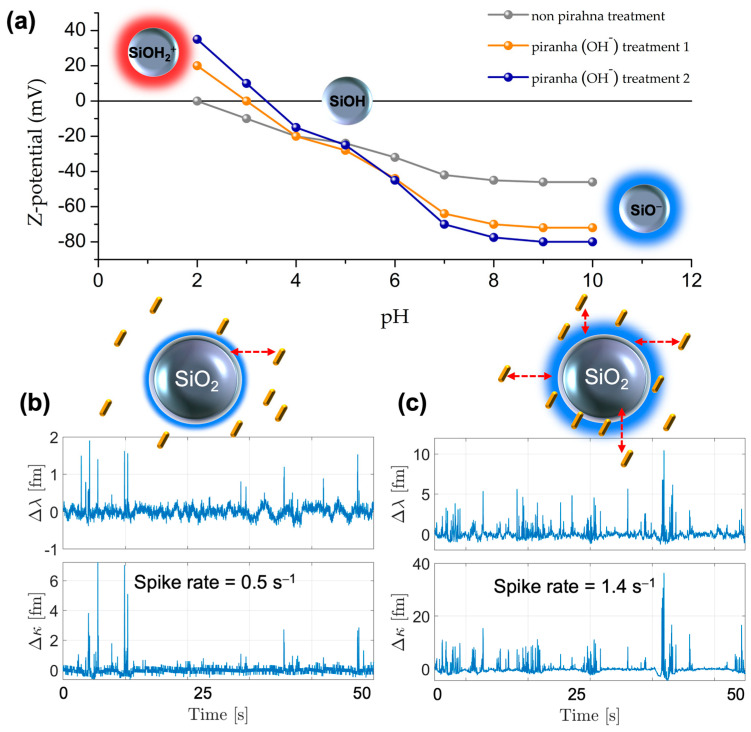
(**a**) Zeta potential versus pH for silica microspheres in aqueous solutions and after activation with alkaline piranha solution. (**b**,**c**) Time series for the variation of wavelength position (Δλ) and linewidth (Δκ) obtained in experiments with the WGM microresonators exposed to colloidal suspensions of 28 pM Au-NR@HS-Si(OH)_n_ (pH = 7 in aqueous solutions). In (**b**), WGM signal traces are shown for non-treated microresonators, and (**c**) shows the WGM signals after alkaline piranha treatment of the WGM microsphere.

**Figure 3 sensors-23-03455-f003:**
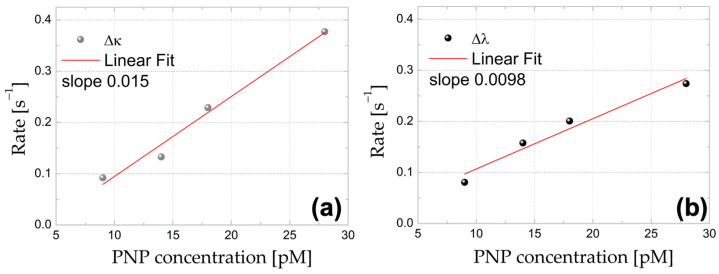
Event rates for (**a**) Δκ and (**b**) Δλ obtained from time series for increasingly higher Au-NR@HS-Si(OH)_n_ concentrations at pH = 7 in aqueous solutions.

**Figure 4 sensors-23-03455-f004:**
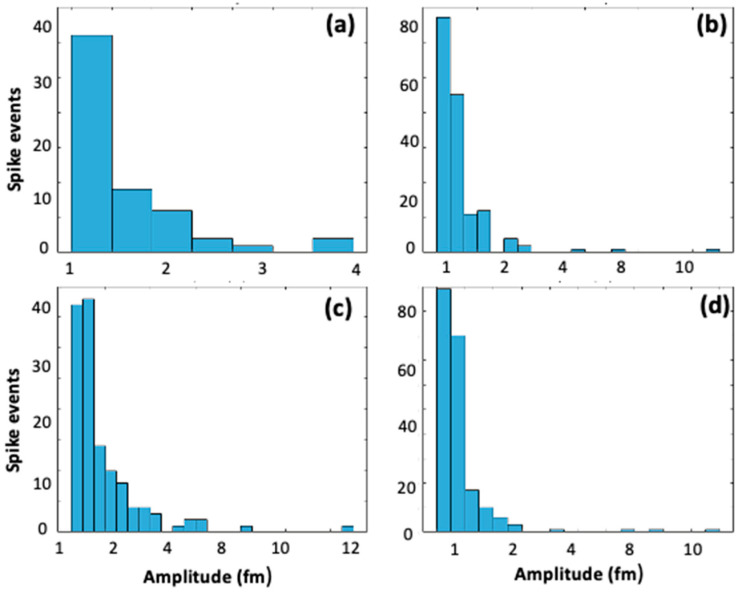
Histograms corresponding to events extracted from time series of Δκ, corresponding to increasing Au-NR@HS-Si(OH)_n_ concentrations explored in Figure 3: (**a**) 9 pM, (**b**) 14 pM, (**c**) 18 pM, and (**d**) 28 pM.

**Figure 5 sensors-23-03455-f005:**
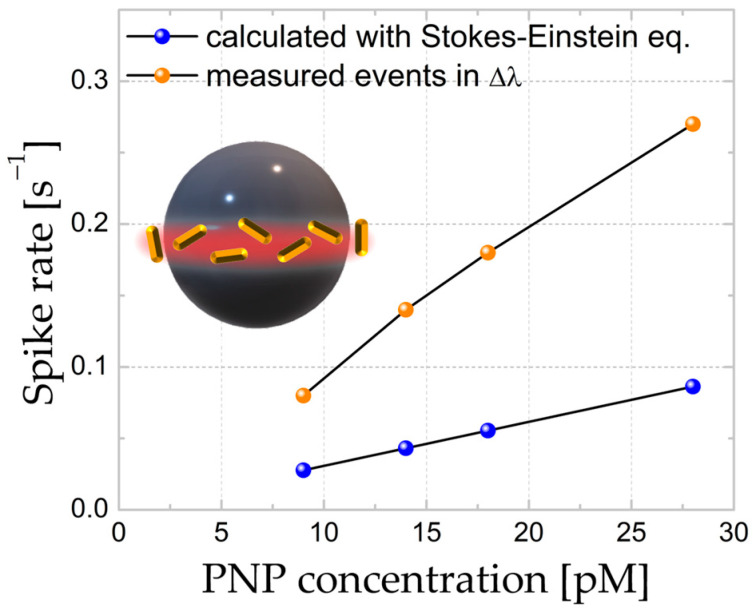
Both spike rates experimentally determined and calculated according to the Stoke–Einstein equation and Fick’s law. Experiments correspond to increasingly higher concentrations of Au-NR@HS-Si(OH)_n_ in pH = 7 aqueous solutions. Exposed area for PNP interaction is 1200 μm^2^, assuming an equatorial section of 5 μm and a radius of 40 μm.

**Figure 6 sensors-23-03455-f006:**
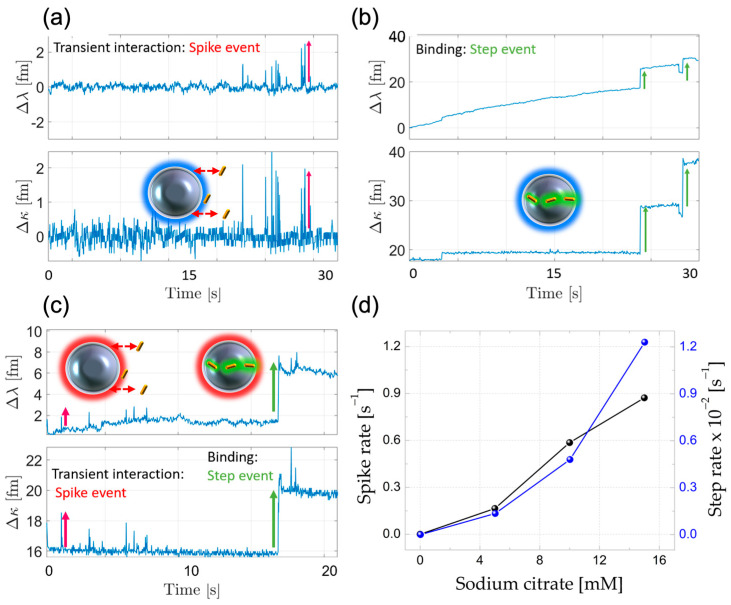
(**a**) WGM sensor traces for citrate-capped AuNR (pH 7, NaCl 10 mM, and 6 pM Au-NR), and (**b**) for 6 pM Au-NR@HS-Si(OH)_n_. (**c**) Both spikes (red arrows) and steps (green arrows) are observed for citrate-capped NR at buffer with pH = 3 and 2 pM Au-NR@HS-Si(OH)_n_. (**d**) Spike and step rate dependences on the concentration of citrate buffer at pH = 3.

**Figure 7 sensors-23-03455-f007:**
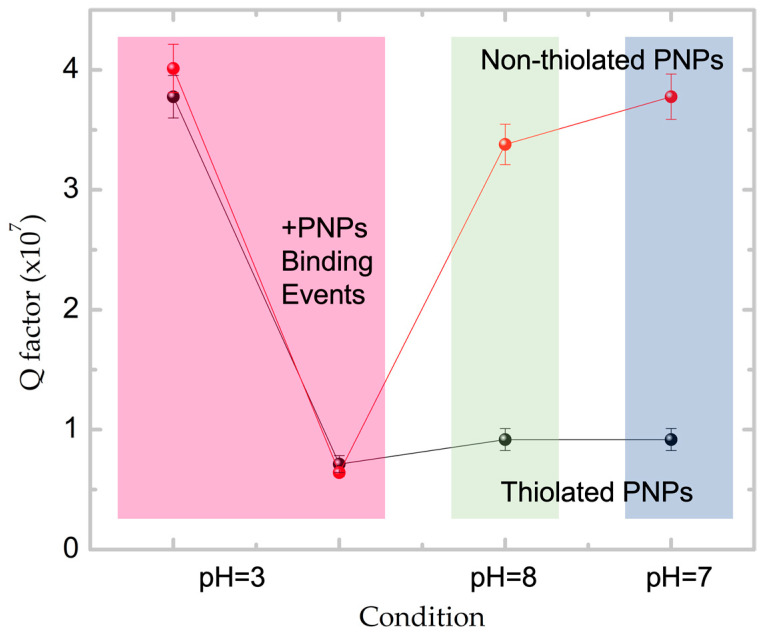
FWHM values corresponding to pH = 3 assembled sensors (both electrostatic (red) and covalent (black) binding) after exposure to aqueous solutions with pH = 8, showing different stabilities.

**Figure 8 sensors-23-03455-f008:**
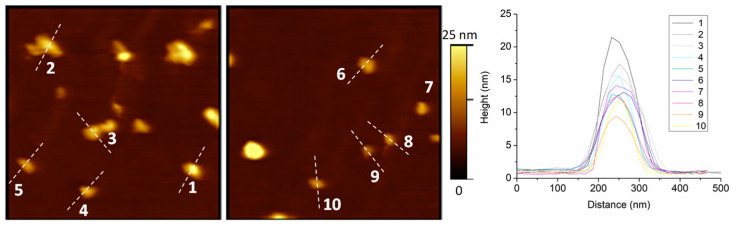
AFM topography images obtained ex situ for samples (2 × 2 µm^2^; height range: 0–25 nm) modified with Au-NR@HS-Si(OH)_n_, and the height for the ten considered nano-objects in the field of view.

**Figure 9 sensors-23-03455-f009:**
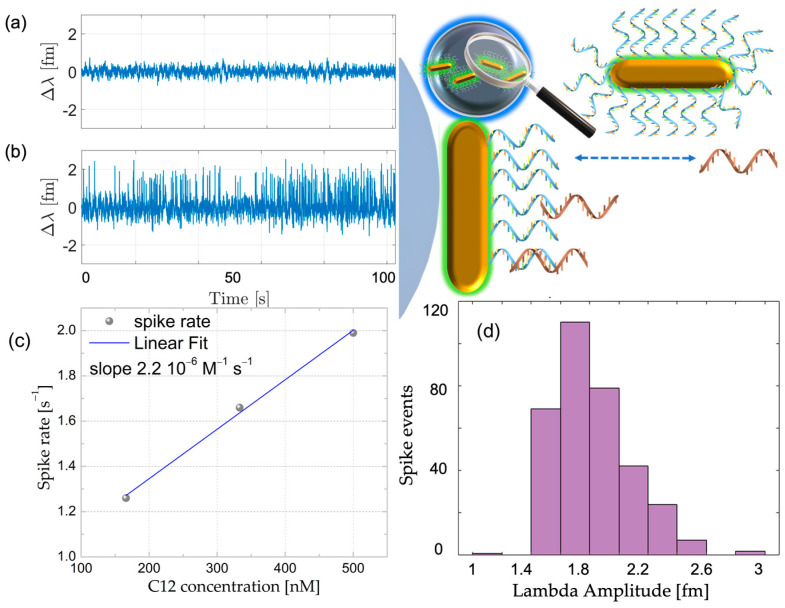
(**a**) Background WGM shift signal for Δλ after 12 h in H_2_O, (**b**) spike signals after adding 166 nM of C12, (**c**) spike rate dependence with C12 full complementary increasing concentration. (**d**) Histograms of the waiting times for events extracted from WGM Δλ shift signal displayed in (**b**) for 166 nM C12.

## Data Availability

Not applicable.

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
