# Peer review of "“Grafting-To” Covalent Binding of Plasmonic Nanoparticles onto Silica WGM Microresonators: Mechanically Robust Single-Molecule Sensors and Determination of Activation Energies from Single-Particle Events"

_sensors, 2023, doi:10.3390/s23073455_

Round 1
Reviewer 1 Report
Mariana P. Serrano et al. had reported an interesting work entitled “Grafting-to covalent binding of plasmonic nanoparticles onto silica WGM microresonators: mechanically robust single-molecule sensors and determination of activation energies from single-particle events”. For publication, the following important questions need to be addressed.
1. The authors did not provide a detailed interpretation of Fig. 3. What does linear increase mean? What does the larger slope of the event rates for Δκ mean?
2. The authors should provide an interpretation for the histograms in Figure 4. Please unify the x-axis range in Figure 4 to avoid misunderstandings for the reader. Please increase the font size of the characters in Figure 4 for readability.
3. The authors estimated the deviation between the experimental and calculated values in Fig. 5. For the understanding of the reader, the authors should provide a more detailed explanation of the causes of the deviation. (Possible questions) (1) What are the probable causes of the spike rate increase? (2) What realistic factors are difficult to include in calculations?
Reviewer 2 Report
This manuscript reports the realization of an innovative biosensor that exploits the properties of gold nanoparticles. To this end, gold nanorods are selected for their high sensitivity to the change in the surrounding medium. After a suitable biofunctionalization, AuNRs are used to detect complementary oligonucleotides. After that, silica microspheres are used as micro resonator-based platforms. The manuscript is fascinating, and the experiments are well-detailed. However, several concerns need to be addressed:
- The spectral characterization of the step-by-step functionalization needs to be reported. It is mandatory to show the plasmonic shift and ensure that AuNRs preserve the plasmonic shapes.
- The AFM topography is not sufficient. I suggest adding high-resolution TEM
- A similar approach has already been reported by Petronella and coworkers (https://pubs.rsc.org/en/content/articlelanding/2022/EN/D2EN00564F). I suggest discussing the differences between the two systems.
- It would be exciting to discuss if the realized systems have intriguing photo-thermal properties. I suggest looking at the work published by N. Halas and coworkers, F. Frantellizzi and coworkers, A. El-Sayed and coworkers.
- The conclusion paragraph needs more details and emphasis.
Reviewer 3 Report
The authors have presented a novel and feasible procedure for covalent attachment of PNPs onto silica WGM microresonators.
Is that possible to characterize such system after “grafting-to” approach? How to determine the “grafting-to” products?
As the title mentions "plasmonic", is that the present experiments utilize this advantage?
In addition, for the experimental results, could the authors make comparasions with reported systems to show the advatages.
Round 2
Reviewer 1 Report
The revised version has been deemed suitable for publication in Sensors.
Author Response
We thank the reviewer for her/his helpfull suggestions, and we think the overall quality of the paper was enhanced after revisions made.
Reviewer 2 Report
In my opinion, points 2 and 4 need to be addressed.
Author Response
Please see document attached.
